# Mass Spectrometry Imaging of Flavonols and Ellagic Acid Glycosides in Ripe Strawberry Fruit

**DOI:** 10.3390/molecules25204600

**Published:** 2020-10-09

**Authors:** Hirofumi Enomoto

**Affiliations:** 1Department of Biosciences, Faculty of Science and Engineering, Teikyo University, Utsunomiya 320-8551, Japan; enomoto@nasu.bio.teikyo-u.ac.jp; 2Division of Integrated Science and Engineering, Graduate School of Science and Engineering, Teikyo University, Utsunomiya 320-8551, Japan; 3Advanced Instrumental Analysis Center, Teikyo University, Utsunomiya 320-8551, Japan

**Keywords:** mass spectrometry imaging, strawberry fruit, flavonols, ellagic acid glycosides, matrix-assisted laser desorption ionization, matrix vapor deposition/recrystallization

## Abstract

Flavonols and ellagic acid glycosides are major phenolic compounds in strawberry fruit. They have antioxidant activity, show protective functions against abiotic and biotic stress, and provide health benefits. However, their spatial distribution in ripe fruit has not been understood. Therefore, matrix-assisted laser desorption/ionization (MALDI)-mass spectrometry imaging (MSI) was performed to investigate their distribution in fruit tissues. Using strawberry extract, five flavonols, namely, three kaempferols and two quercetins, and two ellagic acid glycosides, were tentatively identified by MALDI-tandem MS. To investigate the tentatively identified compounds, MALDI-MSI and tandem MS imaging (MS/MSI) analyses were performed. Kaempferol and quercetin glycosides showed similar distribution patterns. They were mainly found in the epidermis, while ellagic acid glycosides were mainly found in the achene and in the bottom area of the receptacle. These results suggested that the difference in distribution pattern between flavonols and ellagic acid glycosides depends on the difference between their aglycones. Seemingly, flavonols play a role in protective functions in the epidermis, while ellagic acid glycosides play a role in the achene and in the bottom side of the receptacle, respectively. These results demonstrated that MALDI-MSI is useful for distribution analysis of flavonols and ellagic acid glycosides in strawberry fruit.

## 1. Introduction

Either fresh, frozen, or as a processed food, such as yogurt or jam, strawberries (*Fragaria × ananassa*) are the most widespread and consumed berries around the world because of their attractive appearance, delicious taste, and reputed health-related properties [1,2,3,4]. Strawberries are rich in nutritious compounds including sugars, organic acids, and vitamins, as well as a wide range of non-nutritious phytochemicals, especially polyphenols, among which, flavonoids containing flavonols, phenolic acids, lignans, and tannins are the major phenolic compounds found in this fruit [1,2,3,4,5]. Some of the most representative and important flavonols in strawberry fruit are kaempferol and quercetin derivatives, while free ellagic acid and its derivatives are the representative ellagitannins [1,2,3,4,5]. Phenolic compounds containing these flavonols and ellagitannins are well-known for their antioxidative activity potential, whereby they play an important role in plant protection against biotic and abiotic stress factors [1,2,3,4,5]. In addition, these flavonols and ellagitannins show anti-inflammatory activity and possess direct and indirect antimicrobial, anti-allergy, and anti-hypertensive properties, as well as the capacity for inhibiting the activity of some enzymes and receptors, thereby preventing oxidative stress-related diseases [1,2,3,4,5]. Therefore, in addition to other classes of metabolites, such as vitamins, these flavonols and ellagitannins are also considered to be associated with the effects of strawberry fruit consumption on preventing oxidative stress- and inflammation-related diseases, such as metabolic syndrome, cardiovascular diseases, certain types of cancer, and even neurological diseases [1,2,3,4,5,6,7]. Due to the important roles played by these phenolic compounds in plant physiology and human health, understanding their distribution in strawberry fruit tissues is valuable for discerning what strawberry fruit tissues are important for the biological activities described above. However, the spatial distribution of flavonols and that of free ellagic acid, and its glycosides in the body of ripe strawberry fruit, has not been fully determined.

Modern separation techniques, such as liquid chromatography (LC)-mass spectrometry (MS), are useful for qualitative and quantitative analysis of natural substances containing phenolic compounds [8,9,10,11] and have also been used to investigate their spatial distribution by dissecting biological samples into different tissues [11]; however, the resolution is dependent on the accuracy of sampling and tissue differentiation. To compensate for this shortcoming, mass spectrometry imaging (MSI), an emerging technique, has been used to simultaneously investigate the spatial distribution of biomolecules in plant tissues, such as phenolic compounds, without requiring antibodies, staining, or complicated preliminary procedures [12,13,14,15,16,17]. Recently, MSI was adapted for identifying and quantitating various metabolites in foods and plants. This enabled visualization at microscopic resolution using soft ionization techniques, mainly matrix-assisted laser desorption/ionization (MALDI) [12,13,14,15,16,17,18,19,20,21,22,23,24,25,26] or desorption electrospray ionization [27,28,29]. In MALDI-MSI, peak interferences derived from the matrix often occurred for low-mass molecules. In such case, mass spectrometry with high-mass resolving power, or MALDI-tandem MS imaging (MS/MSI), is often useful to discriminate metabolites from matrix-derived peaks [20]. In a previous study, we used MALDI-MSI to visualize and analyze the distribution of anthocyanins and flavan-3-ols, major flavonoids in ripe strawberry fruit [22,23].

In this study, using ripe ‘Tochiotome’ strawberry fruit (a popular strawberry in Japan), we aimed to identify flavonols and ellagic acid glycosides using MALDI-MS/MS analysis. Subsequently, we attempted to visualize the spatial distribution of the tentatively identified flavonols and ellagic acid glycosides using MALDI-MSI analysis. The different spatial distribution of flavonols and ellagic acid glycosides seemingly depends on the difference between their aglycone moieties.

## 2. Results and Discussion

### 2.1. Identification of Flavonols and Ellagic Acid Glycosides in Strawberry Extracts

Several studies using LC-electrospray ionization (ESI)-MS/MS analyses have reported the presence of several types of flavonols, free ellagic acid, and ellagic acid glycosides in strawberry fruit [8,9,10,11]. Table 1 shows the chemical formula, exact mass, and exact *m*/*z* value of the [M − H]^−^ ion for kaempferol, quercetin, and ellagic acid [8,9,10,11], which are the aglycone moieties of flavonols and ellagic acid glycosides analyzed in this study. We analyzed these phenolic compounds in the negative ion mode because they have been commonly analyzed in that same mode by LC-ESI-MS and MALDI-MS [8,9,10,11,21,25]. First, we prepared the strawberry crude extract and performed MALDI-MS analysis using 1,5-diaminonaphtalene (DAN) as a matrix. In the MALDI-MS spectrum at *m*/*z* 280–310 and 440–600, several peaks corresponding to flavonols as well as to ellagic acid and its glycosides were detected (Figure 1A) [8,9,10,11,21,25].

To investigate whether the peaks might truly be attributed to flavonols and ellagic acid glycosides, we performed MALDI-MS/MS analysis for these peaks as precursor ions. In each MS/MS spectrum (Figure 1B–F), other ion peaks were also detected as precursor ions, as the *m*/*z* window for MS/MS analysis was set at the precursor ion *m*/*z* ± 3 Da. An intense peak corresponding to their aglycones was detected in the MS/MS spectra of the flavonols and ellagic acid glycosides precursor ions by LC-ESI- and MALDI-MS/MS [8,9,10,11,21]. The MS/MS spectrum of the *m*/*z* 447.1 ion indicated an intense ion at *m*/*z* 285.0, which corresponded to the kaempferol aglycone (Table 1), and a neutral loss of 162.1 Da, corresponding to the loss of a hexose moiety (Figure 1B). The major hexose moiety of kaempferol in strawberry fruits is reportedly glucose [4,8,9,10,11]. In addition, an intense ion was detected at *m*/*z* 301.0, corresponding to quercetin and ellagic acid aglycones (Table 1), and a neutral loss of 146.1 Da, which corresponded to the loss of a deoxyhexose moiety. The presence of ellagic acid deoxyhexoside but not quercetin deoxyhexoside has also been reported [4,8,9,10,11]. Thus, we tentatively identified the *m*/*z* 447.1 ion as a mixture of kaempferol glucoside and ellagic acid deoxyhexose [M − H]^−^ ions (Table 2). In turn, the MS/MS spectrum of the *m*/*z* 461.1 ion indicated an intense ion at *m*/*z* 285.0, which corresponded to the kaempferol aglycone (Table 1), and a neutral loss of 176.1 Da, which corresponded to the loss of a glucuronic acid moiety (Figure 1C); thus, we tentatively identified the *m*/*z* 461.1 ion as the kaempferol glucuronide [M − H]^−^ ion (Table 2) [4,8,9,10,11]. Meanwhile, the MS/MS spectrum of the *m*/*z* 463.1 ion indicated an intense ion at *m*/*z* 301.0 (Figure 1D), which corresponded to the quercetin and ellagic acid aglycones (Table 1), and a neutral loss of 162.1 Da, which corresponded to the loss of the hexose moiety. Several researchers have reported the presence of both the quercetin glucoside and the ellagic acid hexoside in strawberry fruit [4,8,9,10,11]; thus, we tentatively identified the *m*/*z* 463.1 ion as a mixture of quercetin glucoside and ellagic acid hexoside [M − H]^−^ ions (Table 2). In turn, the MS/MS spectrum of the *m*/*z* 477.1 ion indicated an intense ion at *m*/*z* 301.0 (Figure 1E), which corresponded to quercetin and ellagic acid aglycon [M − H]^−^ ions (Table 1), and a neutral loss of 176.1 Da, which corresponded to the loss of the glucuronic acid moiety. Furthermore, the presence of quercetin glucuronide has been reported only in strawberry fruit [4,8,9,10,11]; thus, we tentatively identified *m*/*z* 477.1 as quercetin glucuronide [M − H]^−^ ion (Table 2). Additionally, the MS/MS spectrum of the *m*/*z* 593.1 ion indicated an intense ion at *m*/*z* 285.0 (Figure 1F), which corresponded to the kaempferol aglycone (Table 1), and a neutral loss of 308.1 Da, which, according to the literature [4,8,9,10,11], corresponded to the loss of a coumaroyl glucose moiety; thus, we tentatively identified *m*/*z* 593.1 as kaempferol coumaroyl glucoside [M − H]^−^ ion (Table 2). Finally, peaks at *m*/*z* 285.0 and 301.0, corresponding to kaempferol, quercetin, and ellagic acid aglycone [M − H]^−^ ions (Table 1), were also detected in the mass spectrum (Figure 1A).

In general, kaempferol and quercetin are present as glycosides in strawberry fruit, whereas ellagic acid is present both in free form and as glycosides. Although anthocyanidins, as flavonols, are generally present as glycosides in strawberry fruit, anthocyanidin aglycones were also detected in a previous study using MALDI-MSI for the analysis of anthocyanin in strawberry fruit [22]. The detection of a peak corresponding to kaempferol may be due to the in-source fragmentation of its glycosides [22,23]. Furthermore, the peak at *m*/*z* 301.0 may contain free ellagic acid, as well as detached ellagic acid and quercetin aglycones from their glycosides that resulted from in-source fragmentation [22,23]. In addition to the flavonols and ellagic acid glycosides reported herein (Table 2), the presence of other flavonols or ellagic acid glycosides, such as ellagic acid pentoside, as major components in strawberry fruit are reported in the literature [4,8,9,10,11]; however, the corresponding peak was not detected, a finding that might be explained by the difference in strawberry cultivar analyzed in this study.

### 2.2. MALDI-MSI of Flavonols and Ellagic Acid Glycosides in Strawberry Fruit

The flesh of the strawberry fruit is a swollen receptacle, i.e., a false fruit, and the seeds or achenes located on the surface of the receptacle are the true fruit, collectively referred to as the strawberry fruit [11]. To investigate the distribution of the tentatively identified flavonol and ellagic acid molecular species in the strawberry fruit (Table 2), we performed MALDI-MSI analysis. Except for the kaempferol coumaroyl glucoside [M − H]^−^ ion (*m*/*z* 593.1), the peaks corresponding to most of the tentatively identified flavonol and ellagic acid molecular species in strawberry crude extract by MALDI-MS/MS analysis were also detected by MALDI-MSI (Figure 2A). Figure 2B is an optical image of a longitudinal section of a strawberry fruit. Red pigment is present predominantly in the epidermis, and less abundantly in cortical and pith tissues, which indicates the presence of other flavonoids, namely, anthocyanins [22]. The peak at *m*/*z* 285.0, tentatively identified as kaempferol aglycone [M − H]^−^ ion, was mainly distributed in the epidermis and in the bottom area of the receptacle (Figure 2C). In turn, the ion peak at *m*/*z* 301.0, tentatively identified as a mixture of free ellagic acid, detached ellagic acid, and quercetin aglycones was mainly distributed in the epidermis and in and around the peduncle (Figure 2D), whereas the ion peak at *m*/*z* 447.1, tentatively identified as kaempferol glucoside and ellagic acid deoxyhexoside [M − H]^−^ ions, was mainly distributed in and around the peduncle (Figure 2E). Additionally, the ion peak at *m*/*z* 461.1, tentatively identified as kaempferol glucuronide [M − H]^−^ ion, was mainly distributed in the epidermis and in the cortical tissue adjacent to the epidermis (Figure 2F). Meanwhile, the ion peak at *m*/*z* 463.1, tentatively identified as quercetin glucoside and ellagic acid hexoside [M − H]^−^ ions, was mainly distributed outside the strawberry section (Figure 2G), indicating that peak interference derived from matrix DAN affected the quercetin glucoside and ellagic acid hexoside [M − H]^−^ ions [20]. Finally, the ion peak at *m*/*z* 477.1, tentatively identified as the quercetin glucuronide [M − H]^−^ ion, was mainly distributed in the epidermis (Figure 2H).

Fait et al. [11] compared the content of kaempferol coumaroyl glucoside between the achene and the receptacle of strawberry fruit and showed that it was predominantly contained in the achene. In the present study, we analyzed the achene in the strawberry fruit section (Figure 2B), and the kaempferol coumaroyl glucoside was detected in the strawberry fruit extract (Figure 1A,F) but not in the strawberry fruit section. In preparing the strawberry crude extract, 80% aqueous methanol was used as an extraction solvent, whereas no such solvent equivalent was used for the matrix coating on the section because DAN coating was conducted by the sole deposition of its vapor and its subsequent recrystallization in water vapor. Therefore, the discrepancy may be due to the presence or absence of an aqueous methanol extraction step in the case of achenes, because flavonols such as kaempferol glycosides present in achenes might not be detectable when the MALDI-MSI protocol is performed without organic solvent extraction.

### 2.3. MALDI-MS/MSI of Flavonols and Ellagic Acid Glycosides in Strawberry Fruit

MALDI-MS/MSI is useful to discriminate analyte peaks from interference peaks such as matrix or metabolites by reconstructing the ion image using its specific fragment ions [18]. MALDI-MS/MSI analysis was performed to investigate the distribution of each kaempferol glucoside, ellagic acid deoxyhexoside, quercetin glucoside, and ellagic acid hexoside in strawberry fruit (Figure 2E,G and Table 2). Figure 3A shows the strawberry section analyzed by MALDI-MS/MSI. The fragment ions were only measured for each precursor ion at an *m*/*z* value of ±3 Da. Although other fragment ion peaks were also detected, fragment ion peaks corresponding to kaempferol (*m*/*z* 285.0) and ellagic acid (*m*/*z* 301.0) aglycones were clearly detected in the MS/MS spectrum of precursor ion at *m*/*z* 447.1 (Figure 3B). From the reconstructed fragment ion images, we inferred that kaempferol glucoside (*m*/*z* 447.1 → 285.0) was mainly distributed in the epidermis (Figure 3C), whereas ellagic acid deoxyhexoside was mainly distributed in the achene and in the bottom area of the receptacle containing the peduncle (Figure 3D). Similarly, the fragment ion at *m*/*z* 301.0, corresponding to quercetin and ellagic acid aglycones, was detected in the MS/MS spectrum of the precursor ion at *m*/*z* 463.1 (Figure 3E). From the fragment ion image (*m*/*z* 463.1 → 301.0), the quercetin glucoside and the ellagic acid hexoside were mainly distributed in the epidermis, the achene, and the peduncle (Figure 3G), indicating that these peaks can be discriminated by MALDI-MS/MSI from the peak derived from the DAN matrix. In turn, the fragment ion image at *m*/*z* 285.0, corresponding to kaempferol, was also reconstructed because the fragment ion at *m*/*z* 285.0 was likewise detected in the MS/MS spectrum (Figure 3E). The fragment ion image (*m*/*z* 461.1 → 285.0) suggested that kaempferol glucuronide was mainly distributed in the epidermis (Figure 3F). However, the fragment ion image was different from the precursor ion image reconstructed by MALDI-MSI (Figure 2F), indicating that the peak corresponding to the kaempferol glucuronide can also be discriminated from the peak derived from other metabolites present in the cortical tissue adjacent to the epidermis.

In the present study, we tentatively identified several flavonol and ellagic acid molecular species by MALDI-MS/MS analysis, namely, kaempferol, quercetin, and ellagic acid glycosides. Subsequently, we visualized the tentatively identified flavonols and ellagic acid glycosides by MALDI-MSI and MS/MSI and showed their characteristic distribution in the strawberry fruit section. 

Recently, strawberry extracts containing free ellagic acid and its glycosides and flavonols have been used as ingredients in functional foods and dietary supplements in combination with other colorful fruits, vegetables, and herbal extracts [4]. The information on the spatial distribution of flavonols and free ellagic acid and its glycosides obtained in the present study will surely contribute to developing adequate procedures for preparing strawberry extracts enriched with these phenolic compounds.

The kaempferol glucoside and the kaempferol glucuronide were mainly distributed in the epidermis (Figure 3C,F), whereas the kaempferol aglycone was mainly distributed in the epidermis and in the bottom area of the receptacle (Figure 2C). Kaempferol aglycone may be generated from its glycosides by in-source fragmentation [22,23]. Therefore, this discrepancy might be due to the presence of other metabolites close to the *m*/*z* value for the kaempferol [M − H]^−^ ion in the bottom area of the receptacle.

We were not able to visualize the quercetin glucoside and the ellagic acid hexoside separately; they were mainly distributed in the achene, the epidermis, and the bottom area of the receptacle (Figure 3G). However, the quercetin glucuronide was mainly distributed in the epidermis (Figure 2H), while the ellagic acid deoxyhexoside was mainly distributed in the achene and in the bottom area of the receptacle (Figure 3D). A previous analysis of anthocyanins by MALDI-MSI in strawberry fruit [22] found that the different distribution patterns among anthocyanins depended on the type of aglycone moiety. Flavonols containing quercetin glycosides are biosynthesized from the same precursor as anthocyanins, i.e., dihydroflavonol, by the flavonoid biosynthetic pathway [30]. Based on my knowledge of these established facts, I speculated that the quercetin glucoside was mainly distributed in the epidermis, whereas the ellagic acid hexoside was mainly distributed in the achene and in the bottom area of the receptacle.

Phenolic compounds containing flavonols, free ellagic acid, and ellagic acid glycosides play an important protective role in strawberry fruit against pathogen attack and exposure to UV light [10]. Therefore, they are likely allocated to each plant tissue to exert these protective functions [23,24]. Thus, given their distribution patterns, flavonols seemingly play these protective roles in the epidermis, while free ellagic acid and its glycosides play the same roles in the achene and in the bottom of the receptacle.

We were not able to visualize either quercetin, ellagic acid (Figure 2D) or their glycosides (Figure 3G) individually, as the difference in *m*/*z* value among them is only 0.04 Da (Table 1), which was below the mass-resolving power of the mass spectrometer. Nonetheless, the visualization of each quercetin, ellagic acid, and the corresponding glycoside is necessary for further elucidating their biological functions not only in plants but also in animals and humans. At present, mass spectrometers provided with higher mass-resolving power such as the quadrupole-TOF, orbitrap, or Fourier transform-ion cyclotron resonance types have been utilized in MALDI-MSI [29,31]. These types of mass spectrometers may be useful for individually visualizing each quercetin, ellagic acid, and their glycosides.

## 3. Materials and Methods

### 3.1. Reagents

Carboxymethyl cellulose (CMC) sodium salt, methanol, and water were purchased from Wako Chemicals (Tokyo, Japan). α-Cyano-4-hydroxycinnamic acid (CHCA) and DAN were purchased from Tokyo Kasei Co. (Tokyo, Japan). Indium-tin-oxide (ITO)-coated glass slides (100 Ω without MAS coating) were purchased from Matsunami Glass (Osaka, Japan). Peptide calibration standards containing angiotensin II were purchased from Bruker (Billerica, MA, USA). All reagents and solvents used in this study were of analytical grade.

### 3.2. Strawberry (Fragaria × Ananassa Duch.) Fruit Samples

‘Tochiotome’ strawberry fruit were cultivated at the Strawberry Research Center (Tochigi, Japan). Five ripe strawberries were harvested, frozen immediately after harvest, and stored at −80 °C until use.

### 3.3. Extraction of Polyphenols from Strawberry Fruit

Extraction of polyphenols was performed as previously reported [22,23], after minor modifications. Fresh strawberries were homogenized in an equal weight of water at 4 °C, and the homogenate solution was freeze-dried. The strawberry powder was suspended in 80% methanol and kept under continuous shaking at 20 °C for 2 h. The solution was centrifuged at 5000 rpm for 10 min, and the supernatant was collected as polyphenol crude extract for analysis by MALDI-MS/MS to identify the constituent molecular species of flavonols and ellagic acid glycosides.

### 3.4. MALDI-MS and MS/MS Analysis of Flavonols in Strawberry Extracts

MALDI-MS and MS/MS analyses were performed as previously described [22,23], after minor modifications. Strawberry extract (1 μL) was mixed with an equal volume of 10 mg/mL DAN in 80% aqueous methanol and transferred onto an ITO-coated glass slide and dried. MALDI-MS analysis was performed using a MALDI time of flight (TOF)/TOF-type instrument (UltrafleXtreme, Bruker) equipped with a 355-nm Nd:YAG laser at a repetition rate of 1000 Hz in negative-ion mode (reflector mode). Ions with *m*/*z* values in the range of 260–600 were measured. The *m*/*z* values were calibrated externally using the exact *m*/*z* values for CHCA [M − H]^−^ ions (*m*/*z* 188.03532) and angiotensin II [M − H]^−^ ions (*m*/*z* 1044.52725) as references.

For MS/MS analysis, the selected precursor ions and the product ions were obtained by UltrafleXtreme in collision-induced dissociation “LIFT” MS/MS mode. The obtained MS/MS spectra were analyzed using the flexAnalysis 3.4 software (Bruker). Sugar moieties of flavonols and ellagic acid glycosides were tentatively identified by their neutral losses and the available literature [8,9,10,11,21,25]. The MS/MS window was set at each precursor ion *m*/*z* value ± 3 Da.

### 3.5. Preparation of Fruit Sections

Fruit sections were prepared using CMC freeze embedding as previously described [22,23]. Briefly, fresh fruit were immersed in 2% CMC and flash-frozen in liquid nitrogen. Subsequently, 100-μm thick longitudinal sections were consecutively prepared using a CM 1860 cryostat (Leica Microsystems, Wetzlar, Germany). The sections were mounted onto ITO-coated glass slides and placed in 50-mL conical centrifuge tubes containing silica gel for drying. The sections were preserved at −80 °C in a deep freezer until MALDI-MSI analysis.

### 3.6. Matrix vapor Deposition/Recrystallization

Matrix vapor deposition/recrystallization was performed as previously described [24], after minor modifications. An ITO-coated glass slide loaded with a frozen section was drawn from the freezer and dried in a vacuum desiccator for 30 min. Matrix vapor deposition was performed using the SVC-700TMSG vacuum deposition system (Sanyu Electron Co., Ltd., Tokyo, Japan). The glass slide was mounted on the sample holder using adhesive tape, and DAN (60 mg) was placed on the matrix holder. The distance between the sample holder and the matrix holder was maintained at 5 cm. After the vacuum pressure in the chamber reached less than 5 × 10^−3^ Pa, DAN was heated at 190 to 220 °C for approximately 30 min, until all the DAN powder had sublimed. One milliliter of 10 mM acetic acid solution was placed in a glass Petri dish (90 mm in diameter), and preheated at 35 °C on a hot plate for 10 min. After matrix vapor deposition, the glass slide was kept at 35 °C in the closed Petri dish for 20 min in order to expose the DAN on the section to the solvent vapors.

### 3.7. MALDI-MSI and MS/MSI Analyses

MALDI-MSI analysis was performed according to the previous study [22,23,24], after minor modifications. The strawberry fruit sections were analyzed using a MALDI-TOF/TOF instrument (UltrafleXtreme, Bruker). Data were acquired using a step size of 250 μm in negative-ion and reflector modes to measure *m*/*z* values in the range of 260–600 with the laser diameter set to medium size. The instrument was calibrated externally using the exact *m*/*z* values for CHCA [M − H]^−^ ions (*m*/*z* 188.03532) and angiotensin II [M − H]^−^ ions (*m*/*z* 1044.52725) as references. The spectra were acquired automatically using the FlexImaging 4.1 software (Bruker). Normalization of spectra based on the total ion current was performed using the same software.

Selected precursor and product ions for MALDI-MS/MSI were obtained by UltrafleXtreme in collision-induced dissociation “LIFT” MS/MS mode. For the analysis of precursor ion at *m*/*z* 447.1, the MS/MS window was set to the precursor ion *m*/*z* ± 3 Da. For the analysis of precursor ion at *m*/*z* 461.1 and 463.1, the MS/MS window was set to the precursor ion at *m*/*z* 462.1 ± 3 Da. Fragment ions but not precursor ions were measured by MS/MS analysis. The FlexImaging 4.1 software was also used to create two-dimensional ion-density maps.

To investigate the spatial distribution of the tentatively identified flavonols and ellagic acid glycosides, nine sections prepared from three different strawberry fruit samples (three sections per one strawberry fruit) were analyzed. The mass spectra and ion images of the tentatively identified flavonols and ellagic acid glycosides in the three sections prepared from the same strawberry fruit, and those in different strawberry fruit samples, showed similar patterns (Figure 2, Appendix A); therefore, only the mass spectrum and ion images of one of the three different strawberry fruit samples are shown as representative data in Figure 2.

## 4. Conclusions

We demonstrated that MALDI-MSI analysis is useful for the visualization of flavonols, free ellagic acid, and ellagic acid glycosides. To the best of our knowledge, this is the first report on the visualization of various types of both flavonols and ellagic acid glycosides in strawberry fruit using MALDI-MSI. Currently, researchers around the world are making great efforts to improve food quality traits in strawberry fruit, such as health-related properties, through breeding. MALDI-MSI analysis and the information reported herein on the spatial distribution of flavonols, free ellagic acid, and ellagic acid glycosides in ripe strawberry fruit should prove useful to these efforts, not only for strawberry fruit but for many other fruits as well.

## Figures and Tables

**Figure 1 molecules-25-04600-f001:**
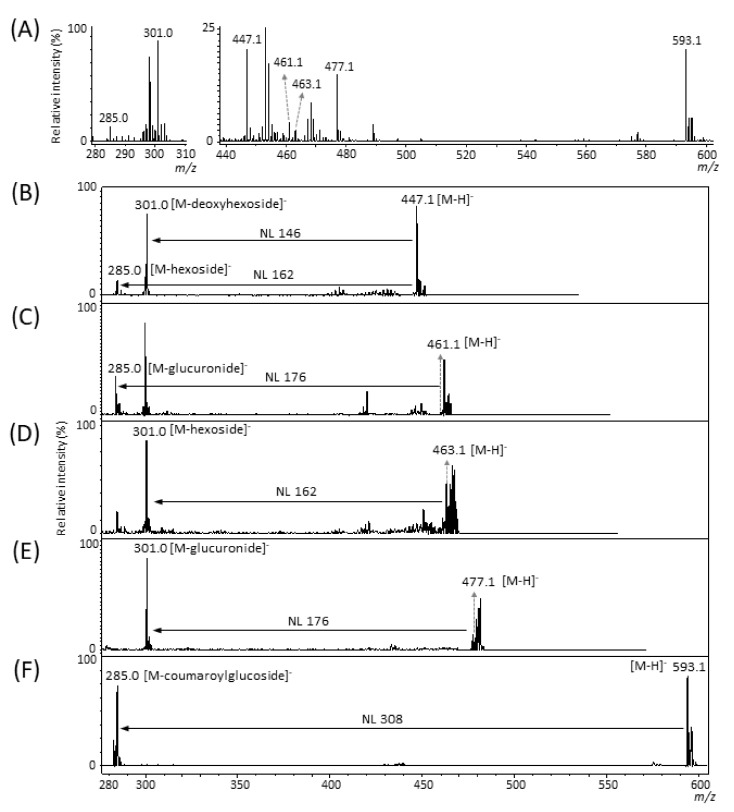
Tandem mass spectra of tentatively identified flavonols and ellagic acid glycosides in ripe strawberry fruit extract obtained by matrix-assisted laser desorption/ionization (MALDI)-tandem mass spectrometry (MS/MS). Mass spectrum of the extract at (**A**) *m*/*z* 280–310 and 440–600 by MALDI-MS. The *m*/*z* values indicate that the peaks were possibly attributable to flavonols and ellagic acid glycosides. MS/MS spectra of precursor ions at *m*/*z* (**B**) 447.1, (**C**) 461.1, (**D**) 463.1, (**E**) 477.1., and (**F**) 593.1. Neutral losses (NL) indicate the losses of sugar moieties.

**Figure 2 molecules-25-04600-f002:**
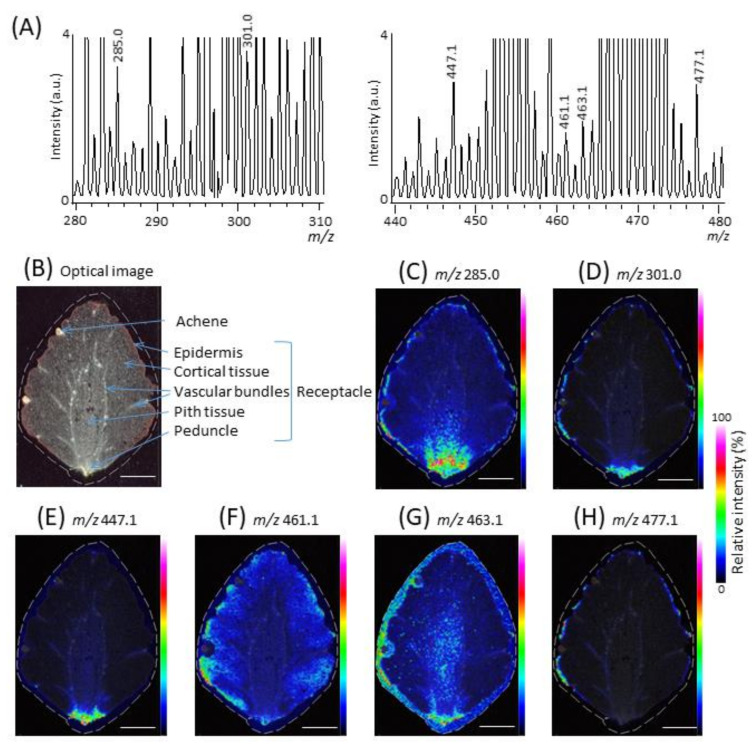
Matrix-assisted laser desorption/ionization-mass spectrometry imaging of the tentatively identified flavonols and ellagic acid glycosides in a ripe strawberry fruit section. (**A**) Mass spectra at *m*/*z* 280–310 and 440–480 of the section. Based on the *m*/*z* values, the peaks were tentatively identified as flavonols and/or ellagic acid glycosides. (**B**) Optical image of the section before measurement. Ion images at *m*/*z* (**C**) 285.0, (**D**) 301.0, (**E**) 447.1, (**F**) 461.1, (**G**) 463.1, and (**H**) 477.1. The dotted white line shows the analyzed region. Scale bar = 5 mm.

**Figure 3 molecules-25-04600-f003:**
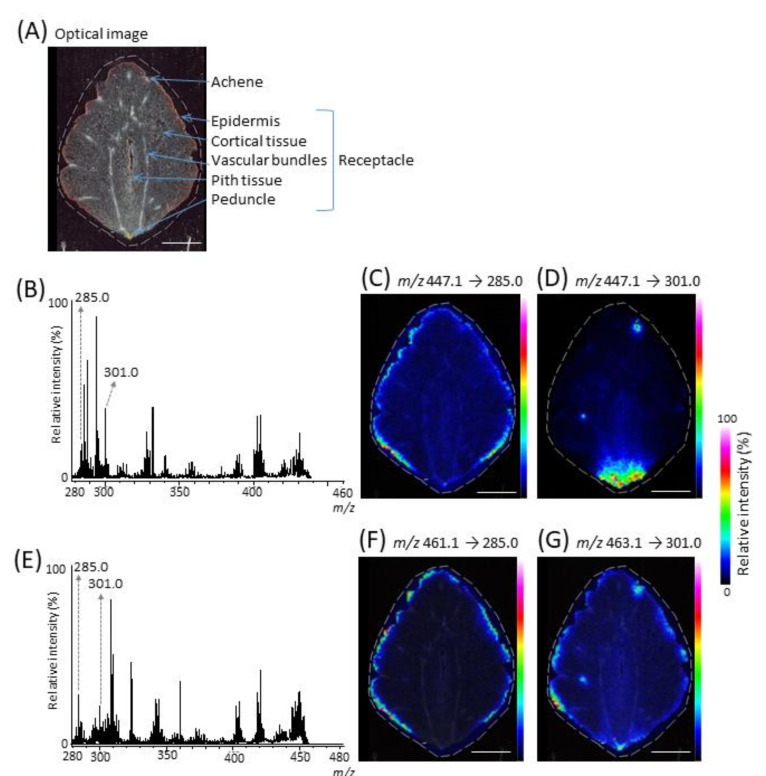
Matrix-assisted laser desorption/ionization (MALDI)-tandem mass spectrometry imaging (MS/MSI) of tentatively identified flavonols and ellagic acid glycosides in a ripe strawberry fruit section. (**A**) Optical image of the section before measurements. (**B**) MALDI-MS/MSI spectrum of precursor ion at *m*/*z* 447.1 and its fragment ion images at *m*/*z* (**C**) 285.0 and (**D**) 301.0. (**E**) MALDI-MS/MSI spectrum of precursor ion at *m*/*z* 461.1 and 463.1, and its fragment ion images at *m*/*z* (**F**) 285.0 and (**G**) 301.0. In the MALDI-MS/MSI analysis, only fragment ions (not precursor ions) were measured. The dotted white line shows the analyzed region. Scale bar = 5 mm.

**Table 1 molecules-25-04600-t001:** Aglycones of tentatively identified phenolic compounds in ripe strawberry fruit.

Aglycone	Chemical Formula	Exact Mass	Exact *m*/*z*, [M − H]^−^
Kaempferol	C_15_H_10_O_6_	286.05	285.04
Quercetin	C_15_H_10_O_7_	302.04	301.04
Ellagic acid	C_14_H_6_O_8_	302.01	301.00

**Table 2 molecules-25-04600-t002:** Flavonols and ellagic acid glycosides tentatively identified in ripe strawberry fruit.

Precursor Ion [M − H]^−^, (*m*/*z*)	Fragment Ion Used for Identification [M − H]^−^, (*m*/*z*)	Tentatively Identified Molecular Species
285.0	*n.d.*	Kaempferol
301.0	*n.d.*	Quercetin and ellagic acid
447.1	285.0	Kaempferol glucoside
447.1	301.0	Ellagic acid deoxyhexoside
461.1	285.0	Kaempferol glucuronide
463.1	301.0	Quercetin glucoside and ellagic acid hexoside
477.1	301.0	Quercetin glucuronide
593.1	285.0	Kaempferol coumaroyl glucoside

Strawberry fruit extract was analyzed by matrix-assisted laser desorption/ionization-mass spectrometry. Identification of molecular species was based on literature [5,8,9,10,11]. *n.d.* means not detected.

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
