# Peer review of "Mass Spectrometry Imaging of Flavonols and Ellagic Acid Glycosides in Ripe Strawberry Fruit"

_molecules, 2020, doi:10.3390/molecules25204600_

Round 1
Reviewer 1 Report
This manuscript“Mass spectrometry imaging of flavonols and ellagic acid glycosides in ripe strawberry fruit” reported an investigation to identify flavonols and ellagic acid glycosides using MALDI-MS/MS analysis. The work is interesting and the manuscript could only be accepted after revision.
- In this work, authors identified several flavonol and ellagic acid molecular species by MALDI-MS/MS analysis, and visualized the identified flavonols and ellagic acid glycosides showed their characteristic distribution in the strawberry fruit section. There are so many interferingsubstance in fruit, how to remove these interference?
- In Figure 2, how many fruit-section were analyzed in this work? Are the results same? The characteristic distribution in the strawberry fruit section was also provided in this manuscript, what is the difference between the interior section and surface?
- The following references about flavonols can be cited in this manuscript.
- Journal of Biomolecular Structure and Dynamics, 2018, 36:13, 3388-3397.
- Journal of Molecular Recognition, 2017, e2606.
- Physics and Chemistry of Liquids (2018), 56(4), 482-495.
Author Response
We thank these reviewers for carefully reading our manuscript and for giving useful and encouraging comments.
Responses for reviewer 1
No. 2: In this work, authors identified several flavonol and ellagic acid molecular species by MALDI-MS/MS analysis, and visualized the identified flavonols and ellagic acid glycosides showed their characteristic distribution in the strawberry fruit section. There are so many interfering substance in fruit, how to remove these interference?
Response: As described in Materials and Methods section, I have not performed any sample pretreatments to remove interfering compounds. The tentatively identified flavonols and ellagic acid glycosides were detected under the experimental conditions described in the revised manuscript.
Several researchers (reference No. 5, 8-11) also reported the presence of these tentatively identified flavonols and ellagic acid glycosides in strawberry fruits. Therefore, the concentrations of these flavonols and ellagic acid glycosides are considered to be relatively higher than other compounds in strawberry fruit. From this thing, I guess that the tentatively identified flavonols and ellagic acid glycosides are not affected by ion suppression derived from other compounds.
No. 3: In Figure 2, how many fruit-section were analyzed in this work? Are the results same? The characteristic distribution in the strawberry fruit section was also provided in this manuscript, what is the difference between the interior section and surface?
Response: I analyzed nine sections prepared from three different strawberry fruit samples (three sections per one strawberry fruit). The ion images of tentatively identified flavonols and ellagic acid glycosides in the three sections prepared from same strawberry fruit, and those in different strawberry fruits showed similar patterns. The ion images of other two strawberry sections were added as Supplementary Figure S1 in the revised manuscript. I corrected the sentences related to these in the revised manuscript (Lines 335–341).
In this study, I showed the different distribution patterns among interior tissues, namely, cortical tissue, vascular bundles, pith tissue, peduncle, and surface, namely, epidermis, and achene.
No. 4: The following references about flavonols can be cited in this manuscript.
Journal of Biomolecular Structure and Dynamics, 2018, 36:13, 3388-3397.
Journal of Molecular Recognition, 2017, e2606.
Physics and Chemistry of Liquids (2018), 56(4), 482-495.
Response: I cited the two of three papers suggested in the revised manuscript, because I confirmed that the two papers are studies on the health related properties of flavonols. Their reference number is 6, and 7.

Reviewer 2 Report
This is a very interesting paper describing MALDI-MS/MS analysis of some flavonols and ellagic acid glycosides in strawberry fruit.
The treated argument is of importance to the field, the used experimental approach and the technical quality are of good quality. The data are correctly presented and discussed by the Authors.
Perhaps I think is more correct to use the sentence ‘tentatively identified compounds’ instead of ‘identified compounds’ in all the manuscript.
In my opinion the manuscript can be accepted for publication with only minor revision as suggested.
Author Response
We thank these reviewers for carefully reading our manuscript and for giving useful and encouraging comments.
Responses for reviewer 2
No. 5: Perhaps I think is more correct to use the sentence ‘tentatively identified compounds’ instead of ‘identified compounds’ in all the manuscript.
Response: I changed the sentence ‘identified compounds’ to ‘tentatively identified compounds’ entire the manuscript.
